# Selection of Bacteriophages to Control In Vitro 24 h Old Biofilm of *Pseudomonas aeruginosa* Isolated from Drinking and Thermal Water

**DOI:** 10.3390/v11080749

**Published:** 2019-08-13

**Authors:** Vanessa Magin, Nathalie Garrec, Yves Andrés

**Affiliations:** 1Centre Scientifique et Technique du Bâtiment, Plateforme AQUASIM, 44300 Nantes, France; 2Institut Mines Télécom Atlantique, Laboratoire de Génie des Procédés Environnement Agro-alimentaire (UMR CNRS 6144), 44300 Nantes, France

**Keywords:** bacteriophages, *Pseudomonas aeruginosa* isolates, planktonic cells, biofilms, stainless steel surfaces, disinfection, viability q-PCR

## Abstract

*Pseudomonas aeruginosa* is an opportunistic pathogen that causes public healthcare issues. In moist environments, this Gram-negative bacterium persists through biofilm-associated contamination on surfaces. Bacteriophages are seen as a promising alternative strategy to chemical biocides. This study evaluates the potential of nine lytic bacteriophages as biocontrol treatments against nine environmental *P. aerginosa* isolates. The spot test method is preliminarily used to define the host range of each virus and to identify their minimum infectious titer, depending on the strain. Based on these results, newly isolated bacteriophages 14.1, LUZ7, and B1 are selected and assessed on a planktonic cell culture of the most susceptible isolates (strains MLM, D1, ST395E, and PAO1). All liquid infection assays are achieved in a mineral minimum medium that is much more representative of real moist environments than standard culture medium. Phages 14.1 and LUZ7 eliminate up to 90% of the PAO1 and D1 bacterial strains. Hence, their effectiveness is evaluated on the 24 h old biofilms of these strains, established on a stainless steel coupon that is characteristic of materials found in thermal and industrial environments. The results of quantitative PCR viability show a maximum reduction of 1.7 equivalent Log CFU/cm^2^ in the coupon between treated and untreated surfaces and shed light on the importance of considering the entire virus/host/environment system for optimizing the treatment.

## 1. Introduction

Antimicrobial control is essentially based on the use of antibiotics in human and animal medicine and on the use of disinfectant products grouped under the name “biocides” for all other uses. The ability of the microbial community to adapt to these different molecules, which are sometimes used in an unreasonable way, the worrying increase in antibiotic resistance, and the lack of suitable solutions to eliminate certain pathogens from our environment led us to develop alternative solutions while considering economic and social issues.

*P. aeruginosa* is an important opportunistic pathogen which can be involved in biofilm-associated contamination of surfaces in moist environments. Once attached to the solid surface, bacterial cells secrete a matrix of extracellular polymeric substances (EPSs) that helps to protect them from external stresses such as the action of disinfectants [1,2,3,4,5]. In the late stage of maturation, biofilms play the role of reservoirs, which could contribute to pathogen dissemination by detachment or tearing under external constraints. Although harmless to healthy people, the presence of *P. aeruginosa* in hospitals and thermal spas is problematic. These bacteria can colonize wounds, or the urinary tract, eyes, ears, or lungs of persons who are immunodeficient, severely impairing their health. So, the robustness of biofilms makes their elimination difficult, despite the existing disinfection protocols in environments where their control is necessary, such as water networks.

Bacteriophages are natural predators, which only infect specific bacteria [6]. A phage can attach to the host bacterial cells by recognizing specific receptors [7]. Following the injection of its genome, the prokaryotic virus can self-replicate by using cellular host machinery and synthesize progeny copies inside the cell. Finally, host cell membrane disruption leads to bacteria lysis and the release of lytic phages, which makes a new cycle of infection possible. Their huge diversity in the environment gives us the opportunity to isolate virulent phages to selectively target pathogenic bacteria without disturbing the normal microflora. Research on phages has been extensive over the last decade and has already shown their effectiveness in treating infectious diseases in medicine [8,9] and in controlling bacterial pathogens in food processing environments as well as in agriculture [10,11,12,13,14]. Thus, bacteriophages might be a promising alternative (or, at least, a supplementary approach) to current non-selective disinfection strategies, which are responsible for material corrosion and by-product dissemination. Moreover, the ability of phages to self-replicate as long as the host is present implies that, in contrast to other biocides, a single dose would be sufficient. Despite their great benefits, there is nevertheless a need for in vitro assays to avoid inappropriate use.

Whereas promising results shed light on the ability of bacteriophages to remove and prevent biofilm growth on different kind of surfaces [15,16,17,18,19,20,21,22], research also highlights complex interactions that strongly depend on the system virus/host/environment studied [23] and on phage/bacteria evolution [24]. In the case of *P. aeruginosa*, the majority of the studies dealing with the efficacy of phages on planktonic cells or biofilms concern clinical isolates [15,20,25,26,27,28,29]. Despite a great diversity in the phages infecting *P. aeruginosa* [30], very few studies have revealed phages that can infect non-clinical isolates [31]. Moreover, in vitro assays are often performed in standard culture media (Luria Bertani broth (LB), Tryptic soy broth (TSB)) and in 24 or 96-well polystyrene plates, far from real conditions. When studies have explored the potential of bacteriophages to remove biofilm from industrial surfaces, they have focused on the most prevalent foodborne pathogens, *E. coli* [32], *Listeria monocytogenes* [33,34,35], *Salmonella* spp. [36], and *P. fluorescens* [37], and to a lesser extent *P. aeruginosa* [38], despite its prevalence on these kind of surfaces.

Therefore, this study investigates the potential of bacteriophages to act as biocontrol agents of planktonic and biofilm states of environmental water networks of *P aeruginosa* isolates. Seven tailed bacteriophages, already known to be specific to *P. aeruginosa*, and two newly isolated ones are screened to analyze their host range and virulence by the spot test method. The most effective ones are then selected to assess their bactericidal pattern on planktonic cells growing in a standardized mineral minimum medium that mimics the commonly encountered nutrient-limiting conditions in natural, moist *P. aeruginosa* environments. To obtain a representative model of infections, 24 h old biofilm contaminations are allowed to develop on stainless steel coupons, a steel alloy characteristic of surfaces encountered in humid environments such as agri-food equipment or thermal water networks. In order to monitor the phage susceptibility of biofilm cells, a reliable detection method is required to quantify the remaining population of viable bacteria. Indeed, the ability of *P. aeruginosa* to switch from a viable to a viable but non-culturable state (VBNC) as an adaptive strategy can lead to underestimation of the number of viable cells by the usual Colony Forming Unit (CFU) count method due to the inherent non-culturability of VBNC cells [39,40]. Here, we use the viability-qPCR (v-qPCR) to estimate the quantity of remaining *P. aeruginosa* cells in biofilm following a single exposure to bacteriophages.

## 2. Materials and Methods

### 2.1. P. aeruginosa isolates: Origin and Culture Conditions

#### 2.1.1. Origin

Nine strains of *P. aeruginosa* were used for this study in the laboratory and are listed in Table 1. This panel included the PAO1 reference strain [41], six environmental strains isolated from drinking water or thermal water networks, as well as two strains of clinical (ST395P) and environmental (ST395E) origin isolated from the same intensive care unit (ICU) of Besançon hospital (Besançon, France). Two strains, the PAO1 reference strain and the ST111 environmental isolate, were provided by the Macromolecular Systems Engineering Laboratory of Marseille (UMR Centre National de la Recherche Scientifique 7255, Marseille, France). The environmental isolates D1 and D2 were provided by the Laboratory of Hydrology-Environment of the Faculty of Pharmaceutical Sciences of Bordeaux (Bordeaux, France). Isolates E1 and E6 came from the same thermal establishment and were isolated from drinking water and thermal water, respectively. 

#### 2.1.2. Strain Culture Conditions and Their Genotypes

Strain typing was performed previously [42] using the Multiple Locus VNTR Analysis (MLVA) method based on the analysis of the following 16 VNTRs: ms77, ms127, ms142, ms172, ms211, ms212, ms213, ms214, ms215, ms216, ms217, ms222, ms223, ms61, ms207 and ms209 (Centre Européen D’expertise Et De Recherche Sur Les Agents Microbiens, La Chapelle sur Erdre, France). 

Each strain was consecutively subcultured on cetrimide selective medium at 30 °C for 48 h and then on plate count-agar (BioMerieux, Craponne, France) at 30 °C for 24 h before being resuspended in liquid Luria Bertani broth (LB) medium or in mineral minimum medium (pasteurized mineral water; 0.02% glucose; 0.05% casamin acid). Bacterial growth was monitored turbidimetrically by measuring absorbance at 600 nm.

### 2.2. Phage Collection 

Part of the panel was kindly provided by the Evolutive Sciences Institute of Montpellier (ISEM, UMR 5554, Montpellier, France). Among them, LKD16, 14.1, LUZ7, and PEV2 were first isolated by Ceyssens et al. [44,45,46]. Seven phages were obtained, with three families of the order *Caudovirales* represented: LKD16, LUZ19, LUZ7, and PEV2 for *Podoviridae*; 14.1, EL for *Myoviridae,* and MP22 for *Siphoviridae*. All have already been the subject of previous studies to confirm their genome double-stranded DNA [47] and were isolated from the *P. aeruginosa* PAO1 or PA14 strains. For some, the host receptors used for infection have been identified. All of these phages were fully sequenced, and corresponding information is available in the European Genome Archive [48] and listed in Table 2. 

#### 2.2.1. Isolation of New Phages

Bacteriophage isolation was performed using a filtration procedure followed by the double-layered agar method described by Adams [53]. Several water samples (municipal sewage, and river water) collected from different places in the department of Pays de La Loire were clarified by centrifugation (5000 rpm for 10 min). The supernatant was filtered with a 0.45 µm syringe filter (Milipore, Merck) to remove the residual organic matter and bacterial cells. An aliquot (1 mL) of filtrate was mixed with 0.1 mL of an overnight culture of the PAO1 host strain and 5 mL of molten top soft nutrient agar (0.7% agar at 56 °C). This was then overlaid on the surface of solidified base nutrient agar (PCA). After overnight incubation at 30 °C, the presence of phages was visualized by small lysis zones. Because each plaque derived from a single phage, they were picked from the plates and amplified three times to ensure the purity of the unique phage isolate. 

Different water samples were screened for the presence of *P. aeruginosa* phages. Using the PAO1 bacterial strain as the host organism, two potentially unknown phages were isolated from river water (La Rochelle, France) and wastewater samples (La Chapelle sur Erdre, France). These bacteriophages produced clear plaques of identical size named B1 and LR1.

#### 2.2.2. Phage Stock Production and Titration

The amplification and extraction steps were previously described by Betts et al. [47]. Briefly, 30 mL of exponentially growing PAO1 culture were infected with one lysis plaque (in the case of the first amplification) or 100 µL of one phage suspension. The culture was placed at 30 °C overnight with soft agitation of 1 min every 30 min. The next day, phage extraction was carried out by adding 10% chloroform (*v*/*v*) and vortexing for 10 s. Bacteria were lysed and thus all phages in the medium were released. The culture was then centrifuged at 13,000 rpm for 4 min. Finally, the supernatant free from chloroform and bacterial debris was filtered before storage at +4 °C for later titration.

Viral titer of the phage lysate was performed according to the double agar overlay technique [54]. Multiple dilutions of the phage isolate were used to find the appropriate dilution to obtain between 20 and 200 plaques. Enumeration of plaque forming units (PFUs), taking into account the dilution factor, gives, finally, the concentration of the initial stock solution (PFU/mL).

#### 2.2.3. Phage Host Range: Spot Assays

The lytic spectra of the phage collection was determined by spot tests [55] on the nine *P. aeruginosa* strains. Briefly, 0.1 mL of target bacteria suspension was mixed with 5 mL of molten top soft nutrient agar (0.7% agar at 56 °C) and spread on the surface of solidified base nutrient agar (PCA). Eight microliters of each phage lysate was spotted onto this bacterial lawn. Drops were allowed to dry before incubation at 30 °C for 16 h. Lytic zones formed on the spotted area were observed and the effectiveness of each individual phage was reported. The experiment was repeated three times for all strains and for each phage. A picture of one spot assay on all stains is presented in Figure A1.

#### 2.2.4. Efficiency of Infection

Primarily, the spot assays above were used to assess the bactericidal activity of all phages, but they also provided the selection of the most sensitive strains. However, a complete lysis zone cannot be considered evidence for productive phage infection, because the lytic zone obtained with a high lysate concentration may be due to bacteriocins or to lysis from without [56]. Serial dilutions of each phage lysate (amplified with PAO1 strain; see part 2.2.2) were thereby performed before eight microliters of each one was spotted onto the agar plate surface containing the targeted host. Results are provided to allow observations of the individual lysis plaques to be made.

Each virus was tested on the susceptible bacterial strains, which were efficiently lysed through the previous spot tests. Assays were done in duplicate for each dilution. After 16 h of incubation at 30 °C, the drops area was analyzed and the number of plaques was enumerated. Efficiency Of Plating (EOP) values were calculated for each phage using the strain PAO1 as a reference [57,58]. Bacteriophage infectivity was categorized on the basis of the EOP directly linked to the number of PFUs. Dark grey squares indicate a “high production” of viruses (EOP = 1 and 1 > EOP ≥0.5), light grey squares underline intermediate EOP (0.5 > EOP ≥ 0.1) and low (EOP < 0.1). Squares stayed white when no lysis was observed (no production). Pictures corresponding to one spot assay result for the four selected isolates challenged by the diluted phage suspensions are presented in Figure A2, Figure A3, Figure A4 and Figure A5. The analysis of these results gave us the opportunity to determine the suitable minimum infectious titer for each virus providing a complete lysis area. This provided us to select the three bacteriophages to use in the study. 

### 2.3. From Spotting to the Planktonic Cells

The bacteriophages B1, LUZ7, and 14.1 were evaluated on the selected strains in liquid medium. An overnight bacterial culture was used to inoculate a mineral minimum medium at OD_600 nm_ = 0.1. This pre-culture of bacteria was placed at 30 °C with stirring in order to obtain the end of the exponential phase of bacterial growth. Experiments were carried out in glass tubes in which 2 mL of the host suspension was mixed with 2 mL of phages (at three different concentrations) to obtain three multiplicities of infection (MOIs) ranging from 10 to 10^−1^. The OD_600 nm_ was measured over 24 h at time intervals of 2 h, and for the last one, after 14 h. Three tubes were monitored for each condition (phages/MOI/strain), and control tubes were performed with a 3 × 2 mL bacterial suspension mixed with an equal volume of mineral minimum medium without phages. Data are reported in graphics, where each point represents the mean ± standard deviation of three tubes.

### 2.4. One-Step Multiplication and Transmission Electron Microscopy (TEM) Observations of B1, 14.1, and LUZ7 Phages

#### 2.4.1. One-Step Multiplication

One-step growth curve studies were performed as described previously [32], with some modifications. A volume of 1 mL of exponential-phase PAO1 culture in LB medium, OD_600_ 0.6 corresponding to approximately 10^9^ CFU/mL, was harvested by centrifugation (5 min, 4000 rpm, 4 °C). An aliquot containing 10^9^ or 10^8^ PFU/mL of phages was added to the bacteria cells, which corresponds to a multiplicity of infection (MOI) of 1 or 0.1. After adsorption for 10 min (30 °C), the mixture was centrifuged, as described above, and the pellet was resuspended in 10 mL of fresh LB at 30 °C under agitation. One aliquot of the supernatant was immediately used to determine the amount of free phage remaining. The reduction of PFU in the supernatants represents the number of phage bound or inactivated by contact with the cells.

Next, duplicate samples of 20 µL were taken aseptically every 5 min over a period of 60 min and were diluted in LB from 10^1^ to 10^8^, allowing phage enumeration to occur. A graph was plotted between the titer of virus and time. The curve obtained at the end of the experiment was used to estimate the burst size and the latent phase of the B1, 14.1, and LUZ7 bacteriophages. 

#### 2.4.2. Transmission Electron Microscopy (TEM)

To visualize the phage morphology, 1 mL of phage lysate was concentrated in two steps via Amicon Ultra-0.5 mL Centrifugal Filter Units 100 K (Merck, Darmstadt, Germany). The purified phage suspension was dripped onto a carbon-formvar coated with a 200 mesh grid and fixed for 1 min, and the extra liquid was removed by tight contact with absorbent paper. Grids were air-dried for 30 s at room temperature before being negatively stained with UranyLess (Delta Microscopie, Mauressac, France), according to the manufacturer’s recommendations. After air drying, grids were examined using a TEM JEM 1010 (JEOL Europe SAS), operating at an acceleration voltage of 80 kV.

### 2.5. Infection of Biofilm Cells Developing on Stainless Steel Coupons

#### 2.5.1. Biofilm Growth System

Stainless steel coupons of rectangular shape (64 mm × 40 mm and 2 mm thickness) corresponding to 51.2 cm^2^, were prepared. Before use, they were cleaned by total immersion in a 70% ethanol bath for 10 min before undergoing sonication for one hour at 60 °C and two successive rinses of 10 min by total immersion in demineralized sterile water at 60 °C. Coupons were then air dried under a microbiological safety station. The effectiveness of the cleaning protocol was evaluated by ensuring the absence of revivable aerobic flora in the last water rinse. For this purpose, the last rinsing water was filtered on a 0.45 µm membrane incubated on PCA medium for 4 days at 30 °C.

Using the same principle as the infection assays in liquid medium, an exponential host bacterial growth culture was used for inoculation at OD_600 nm_ = 0.1 in 3 × 3 reusable plastic jars (Grosseron, Nantes, France) containing 130 mL of mineral minimum medium (Figure 1). One coupon was placed in each jar so that both sides were submerged. Jars were closed but not totally sealed to allow air through. Biofilms were allowed to growth on surfaces for 24 h at 30 °C under static conditions.

#### 2.5.2. Bacteriophage Stainless Steel Surface Disinfection Assays

After biofilm formation, all coupons were washed once to remove all non-adhered bacteria. The coupons were then put into different jars filled with 130 mL of fresh mineral minimum medium. Three of them were immediately sonicated for 5 min to detach cells and to get 24 h biofilm growth controls. Identical phage treatments (approximately 10^10^ PFU/jar) were carried out for 14 h in three other jars, and the last three jars were used as negative controls by adding mineral minimum medium without phages (Figure 1). The six jars were placed again at 30 °C with stirring for 1 min every 60 min.

#### 2.5.3. Titer Monitoring of Bacteriophages

In order to better understand the repartition of phages after the treatments, samples were taken to determine the quantity of planktonic viruses in the mineral minimum medium around stainless steel coupons, but also to estimate the amount that was biofilm-associated. In the first case, samples were collected directly on the liquid medium of the jar after 14 h of phage activity and titered by following the previously detailed protocol. In the second case, at the end of treatment, the liquid medium around the coupons was discarded and replaced by the same volume of medium. The biofilm was then resuspended by sonication (Figure 1), allowing the sampling and titering of biofilm-associated phages. 

Sonication of controls was assessed by adding the same phage titers in each jar (3 × 10^10^ PFU/jar). The titer was then evaluated by the spot test method when jars had not been sonicated and after 1, 3, and 5 min of sonication. 

#### 2.5.4. Quantification of *P.aeruginosa* Population from Biofilms by Viability qPCR, qPCR, and Standard Plate Count (SPC)

The viability PCR (v-PCR) or viability qPCR (v-qPCR) is based on the DNA detection of cells with intact cell/wall membranes, and recently, the use of nucleic acid-binding dye PEMAX^TM^ reagent has allowed the detection and amplification of only the DNA of cells with an active metabolism. Populations were also estimated by standard qPCR and, in the case of untreated jars, by the SPC (standard plate count method).

*Nucleic acid-binding dye in combination with qPCR.* Inhibition of PCR amplification of non-viable cells was performed by the PEMAX™ reagent system (GenIUL, Barcelona, Spain). The procedure was adapted from the manufacturer’s instructions and from previous studies by Nocker et al. [59], Tavernier et al. [60], and Daranas et al. [61].

All coupons (treated or untreated) were rinsed with mineral minimum medium to eliminate phage solution, before being separately sonicated in 130 mL of fresh medium in a new jar. A triple aliquot analysis approach (per jar) was used by taking 3 × 500 µL from each jar in order to detect (Figure A6) (1) viable cells, and (2) the total cell population. The first aliquot (1) was directly treated with a 50 µM PEMAX™ monodose (std. buffer) (GenIUL, Barcelona, Spain). Briefly, the microtubes were placed in the dark at 30 °C for 15 min. Then, the samples were allowed to cool and were photo-activated at 100% for 15 min in a PhAST Blue system (GenIU, Barcelona, Spain). The second aliquot (2) was not treated and was used as the qPCR standard. Controls were carried out by killing bacteria from a third sample before treatment with PEMAX. This procedure was repeated three times per jar, and all aliquots were then stored at −80 °C before DNA extraction. 

#### 2.5.5. DNA Extraction with the NucliSens miniMAG Instrument (NMAG) 

Defrosted samples, PEMAX^TM^ treated or not, were processed using the NMAG system (bioMerieux, Boxtel, The Netherlands) with the NucliSens miniMAG instrument following the instructions provided. DNA was eluted in 50 µL of DNA free water in a shaking incubator at 60 °C (Eppendorf Thermomixer Compact, Copenhagen, Denmark). Additionally, one negative control was included at each round of extraction. Extracts were stored at −80 °C until further PCR analysis.

*Viability qPCR parameters.* Following DNA extraction, the samples were analyzed by qPCR on the Rotor Gene Q thermocycler (Qiagen, Coutaboeuf, France). Amplifications and quantifications were made using the PCR TaqMan QuantiNova Probe kit (Qiagen, Coutaboeuf, France). The two primers and the probe have already been used by authors [62,63,64]. Five microliters of each extracted sample were added to 20 µL of the PCR mixture containing 12.5 µL of PCR TaqMan QuantiNova mix (Table 3), followed by fluorescent data acquisition. Besides the specificity of probe for the detection of the *P. aeruginosa* targeted gene *gyrB*, preliminary tests were accomplished with a melting temperature ramp from 55 to 95 °C at 1 °C/5 s, and the amplicon’s size was 220 bp. Additionally, for each amplification round, negative (RNase-free water) controls were included. Each amplification was performed at least twice per DNA sample.

*Standard curves.* To generate a standard curve, DNA extracted from serially-diluted planktonic *P. aeruginosa* cultures was used for qPCR. The Ct-values (number of cycles) obtained were plotted against the number of viable cells determined by the standard plate count (SPC). The serial dilutions were prepared from a *P. aeruginosa* exponential growth suspension. Cells were diluted from 10^9^ to 10^3^ CFU/mL in sterile water. Two independent biological repeats were included. qPCRs with the TaqMan^®^ probe of the calibration range provided a fluorescence amplification curve for each one of the dilutions. From these amplification curves, points of intersection with the threshold line made it possible to determine the number of cycles (Ct) from which the fluorescence was detected. Means of Ct obtained (*n* = 2) were then plotted against the number of bacteria counted (*n* = 2), enabling us to draw a standard curve of negative correlation: y = −3.9186X + 44.024 with R^2^ = 0.9735 (Table A1).The total amount of cells was determined by qPCR/v-qPCR by interpolating the Ct values from the unknown samples against the corresponding developed standard curve expressed as the equivalent log_10_ CFU per cm^2^ of coupons. 

## 3. Results and Discussion

### 3.1. Phage Spectrum: Selection of Susceptible Strains

Phages were preliminarily amplified to produce sufficient stock lysate. Screening of their host range was then performed. Thus, the infectivity of the seven phages plus two new isolated phages was investigated by the spot method against nine different *P. aeruginosa* strains including the PAO1 reference. Results revealed that on the basis of three replicates, the majority of phages were able to efficiently lyse three environmental pathogenic isolates (Table 4): MLM, D1, ST395E, and not surprisingly, the PAO1 reference strain, which is often used as a phage isolation step. To a lesser extent, some viruses were found to be less effective against D2, ST111, E1, and E6 and totally non-infectious against ST395P. It is interesting to note that LUZ7 and 14.1 were able to infect up to six strains of the collection and B1 and PEV2 infected five of the nine strains tested. The receptors used by LUZ7 and 14.1 phages to adhere to the bacterial surface during infection are lipopolysaccharides (LPS), one of the major components of the Gram-negative bacterial wall. In their study, Silva et al. [7] also highlighted the fact that numerous phages preferentially adhere to the surface of Gram-negative bacteria through these components. With regard to PEV2 phages as well as newly isolated B1 phages, no bacterial receptor is known. 

### 3.2. Effectiveness of Individual Phage: Selection of Virulent Phages

Based on previous experiments, four *P. aeruginosa* strains were selected (MLM, ST395E, PA01 and D1). These strains were challenged with all phages to estimate the effectiveness of each one according to the strain. The spot test method was also used, but standardized phage solutions (10^9^ PFU/mL) were diluted from 10^−1^ to 10^−5^ before being spotted onto the agar lawn. The minimum infectious titer (MIT) was determined for each infectious phage. Results from Figure A2, Figure A3, Figure A4 and Figure A5 are summarized in Table 5 and were used to select three viruses.

The trials showed the same results for the eight phages concerning the PAO1 strain with the smallest MIT (10^6^ PFU/mL). The LUZ7, PEV2, and 14.1 bacteriophages were able to efficiently lyse the MLM and ST395E strains at the same titers, for example, 10^7^ PFU/ mL for LUZ7 and PEV2 and 10^9^ PFU/mL for 14.1. All phages were infectious on the D1 strain, and most efficient ones were LUZ7, LUZ19, PEV2, B1, 14.1, and LDK16 with the MIT estimated at 10^7^ PFU/mL. Based on these results, LUZ7 (*Podoviridae*) and 14.1 (*Myoviridae*) were retained for their effectiveness and in order to represent two of the three *Caudovirales* families. The newly isolated B1 phage was the third candidate selected to assess the susceptibility of *P. aeruginosa* planktonic cells from the MLM, D1, and PAO1 strains. 

### 3.3. Observation of B1 Phages

Transmission electron microscopy was used to observe B1 viral particles. The 14.1 and LUZ7 phages were used as references. Observations show particles with a head and tail measuring approximately 100 nm in length, confirming that phage 14.1 is a member of *Myoviridae* (Figure 2A), which is in accordance with the literature. The PB1 family, to which 14.1 belongs, brings together phages with an icosahedral head of about 70 nm and a non-striated tail of about 50 nm [45]. Pictures taken on the same grid also allowed phages without a tail to be observed (Figure 2A). Similar structures in the surrounding environment suggest that during the resuspension of phages in solution by vortexing before depositing them on the grid, some phages could have been damaged by losing their appendage to leave an outgrowth (see arrow). 

This alteration of particles would be a source of bias during TEM characterization [65]. It also suggests that if *Podoviridae* are less affected, 15% to 40% of *Myoviridae* and between 32% and 76% of *Siphoviridae* can be altered during extraction.

In the case of LUZ7 phages, observations made on the purified extracts show structures with hexagonal forms without tails, measuring approximately 50 to 60 nm, suggesting that they are members of the *Podoviridae* family (Figure 2B). In the literature, LUZ7 assimilated to the family of N4 viruses [46] is also described as a Podoviridae with an icosahedral head of 70 nm. Observations made by Ceyssens et al., however, do not allow the tail to be observed, which, in this family, is considered to be less than 30 nm. In some cases, uranyl ions have a strong affinity with double-stranded DNA, which causes a black color inside the capsid. Similar observations have already been made by E. Bradley [66] in a review dedicated to the study of phage morphology by microscopy. Ackermann and Tiekotter [67] also spoke of positive staining. This artifact is, in most cases, the cause of an underestimation of the size of a viral particle of about 10% to 15%, and does not allow the secondary structures such as the tail or the capsomeres to be observed [67].

Pictures of the B1 unknown extract (Figure 2C) show particles of homogeneous and icosahedral shape of about 60 nm. The absence of a visible caudal appendage during the observations is in favor of viral particles probably belonging to *Podoviridae*. 

### 3.4. Determination of 14.1, B1, and LUZ7 Lytic Parameters

One-step growth curves were perfomed to determine phage multiplication parameters, which are important for characterizing the infection process and establishing their lytic potential. In fact, these parameters allowed us to eliminate the doubt on the existence of other molecular mecanisms which make difficult to distinguish between lysis through phage multiplication, abortive infection, or lysis from without [56]. The one-step growth curves of 14.1 and B1 indicated that the latent period of both phages was approximately 20 min, and the latent period of B1 was very short (10 min) (Table 6). Their estimated burst sizes were 70, 79/90, and 160 phage particles per infected cell, respectively. The high adsorption rates, burst sizes, and short latent periods are in agreement with the lytic activity, which makes them potentially good biocontrol agents.

### 3.5. Liquid Infectious Pattern of LUZ7, B1, and 14.1 in Mineral Minimum Medium 

The efficacy of the three phages, selected above, was assessed by inoculating a culture of *P. aeruginosa* PAO1 and also the three sensitive strains MLM, D1, and ST395E. Suspensions containing the phages were prepared in mineral minimum medium to achieve a multiplicity of infection (MOI, the ratio of virus particles to viable bacterial cells) of 0.1/1/10 in the culture. Phages were introduced after 14 h of bacterial growth, and then cultures were monitored for a further 24 h period. The activity of phages significantly reduced the bacterial population, but the magnitude of these reductions was strain-dependent.

Figure 3, Figure 4 and Figure 5 show that for all controls without phages, absorbance was two-fold higher after 24 h. Conversely, when bacteriophages were infectious, OD_600_ was decreased to near zero. In the case of LUZ7, two patterns were underlined (Figure 3). The populations of the D1 and PAO1 strains were decreased by 90% and 95% respectively, from 8 to 24 h after phage inoculation. However, the MOI which initiated the decline after about 2 h of time course, was the highest. Surprisingly, LUZ7, although selected on the basis of spot tests, did not suppress the growth of the host strains MLM and ST395E at all MOIs tested. 

In the same way, the growth of D1 and PAO1 strains was stopped and then decreased from 2 h after inoculation of phage 14.1 for the highest MOI (Figure 4). Other MOIs (1 and 0.1) also had an effect, but this was delayed in time. Nonetheless, all of the MOIs tested led to the same rate of growth inhibition (approximately 90%) at 10 h post-inoculation. Although the antibacterial effect of phages was significant on the PAO1 and D1 strains, populations seemed to behave differently between 10 and 24 h. Data led to thinking about potential regrowth. 

Compared to LUZ7, the phage 14.1 (at MOI 10) was able to maintain the steady state of the MLM and ST395E populations but not along the whole time course of the infection assay (Figure 4), because after 10 h, the optical density values increased.

The phage activity of B1 was very different from that of LUZ7 and 14.1 (Figure 5). If they had the same infection pattern against the PAO1 strain and D1 isolate, the results showed that B1 could interfere with ST35E and MLM planktonic cells. Concerning the MLM and ST395 cultures, OD_600_ values evolved in the same manner as controls without phages during the first 8 h of phage treatment (Figure 5B,C) and then, surprisingly, phage B1 (at MOI 10) managed to negatively impact the total population, which was reduced by 45% and 70%, respectively (Figure 5B,C). 

Planktonic cells from the reference PAO1 strain, known as a permissive strain, were efficiently eradicated by each of the three viruses, LUZ7, 14.1 and B1 (Figure 3D, Figure 4D and Figure 5D), in liquid. The environmental strain D1 was also sensitive to LUZ7, 14.1 and B1 (Figure 3A, Figure 4A and Figure 5A). While the remaining absorbance for PAO1 could be due to bacterial components of cellular debris, the results show that the regrowth of other strains can be observed more or less rapidly after phage application. This especially concerned MLM and ST35E isolates when infected by LUZ7 and 14.1 (Figure 3B,C and Figure 4B,C). Infections did not suppress multiplication of the bacterial population; nonetheless, their lytic activity maintained a steady state which was probably due to equal lytic and growth rates. However, when the cell multiplication of cultures became higher than the number of dead cells, the optical density proportionally increased according to the MOI used. In the case of 14.1, the growth remained, however, at a lower rate compared to the hosts cell control grown in the absence of bacteriophages (Figure 4B,C). Additional assays were performed by adding a second dose of the virus. No supplemental bactericidal effects were obtained (data not shown) suggesting the selection and multiplication of non-susceptible cells. Similar trends were also observed in earlier studies for *P. aeruginosa* infected with *Podoviridae* or *Myoviridae* phages [20,26] and were associated with the emergence of phage-resistant cells. These phases of steady state bacteria–virus interaction can, to some extent, select natural resistant bacteria or provide the evolution of host cells by developing molecular mechanisms of resistance to bacteriophage infection. Amongst them, Clustered Regularly Interspaced Short Palindromic Repeats (CRISPR), restriction-modification, abortive infection, or the mutation of surface receptors are the most encountered [68]. More recently, the BREX (Bacteriophage Exclusion) system was also described as a system of defense that relies on methylation [69]. 

One of the most important parameters concerns the applied infectious dose. The three MOIs tested in this study led to the same results after 24 h of phage treatment depending on the strain; however, the highest MOI sped up the elimination of bacteria.

Another point of discussion is the lack robustness of the spot test method when screening the lytic activity of phages. Pires et al. [70] also pointed out this aspect in their study. Despite the broad spectra of activity of the four selected phages against clinical *P. aeruginosa* isolates, infection assays on planktonic cells showed that two of them failed to infect their hosts, even at the ideal exponential growth phase. This evidence indicates that phages used for specific antibacterial applications should not be chosen solely on the basis of spot assays. Recently, Xie et al. [71] proposed a microtiter plate-based assay for the determination of the bacteriophage host range and virulence in a high-throughput, 96-well format. This real-time measurement of optical density in the presence of phages would be an alternative host range method with high resolution compared to conventional agar overlay spot assays.

The viruses LUZ7 and 14.1, already known, were chosen to establish the model of the biofilms’ infection. Their effectiveness on D1 and PAO1 planktonic cells makes them good candidates to evaluate their potential on the more complex biofilm structures of these two isolates, respectively. 

### 3.6. Activity of 14.1 and LUZ7 on Biofilms

The following step was to characterize the efficiency of the bacteriophages LUZ7 and 14.1 to remove cells attached to stainless steel surfaces. Previous studies have already highlighted that the antibacterial effectiveness of phages depends partly on the host cells’ physiological state related to the biofilm formation circumstances and, on the other hand, the importance of phage infection modalities [72,73]. Here, juvenile, 24 h old biofilms of D1 and PAO1 *P. aeruginosa* were established on stainless steel coupons immersed in mineral minimum medium. Bacterial populations were quantified by standard plate count (SPC) and/or qPCR/viability qPCR (v-qPCR) before virus application and after. v-qPCR is an accurate specific method of amplification, providing the relative quantification of viable cells only (including VBNC cells) compared to the q-PCR standard, which non-selectively amplifies the targeted sequences of all bacteria (dead, viable, and VBNC) [74,75]. Therefore, the difference between these two types of data is one way to gain information about the repartition state of metabolic cells (between dead cells and viable + VBNC cells). Previously, the use of v-qPCR was also applied to monitor the population dynamics of five species in oral biofilms [76], to quantify *P. aeruginosa* in mono- and multispecies biofilms following exposure to various antibiotics used to treat cystic fibrosis [60], and recently, to monitor viable cells of the biological control agent *Lactobacillus plantarum* PM411 on aerial plant surfaces [61]. 

The viable SPC, qPCR, and v-qPCR results for 24 h old biofilms were respectively 7.40, 7.31 ± 0.23, and 7.12 ± 0.33 Log UFC/cm^2^ of the coupon for PAO1 and 8.20, 7.44 ± 0.4, and 7.18 ± 0.39 Log CFU/cm^2^ of the coupon for D1 (Table 7). Data obtained by qPCR and v-qPCR were not significantly different in the two cases and suggested a homogenous physiological state of cells within the juvenile biofilm. The SPC data were identical to the qPCR results and confirmed that they were mainly composed of viable cells. SPC was also applied to enumerate viable cells of the 24 h + 14 h biofilm, which corresponded to the untreated biofilms. The results show that the populations of the 24 h and 24 h + 14 h biofilms were not significantly different (Table 7 and Table 8). Pires et al. [70] also showed that the quantity of cells in a 24 h old biofilm of the PAO1 strain, growing in 24-well microplates, was identical to that of a 48 h biofilm and was close to 8 Log CFU/cm^2^.

Then, viable cells were assessed 14 h after the addition of phage 14.1 or LUZ7 against PAO1 and D1 24 h-old biofilms by v-qPCR. Data were then compared to the non-treated controls that received no phage, (Figure 6 and Figure 7) (Table 8). The four histograms also showed that there is no significant difference between the standard qPCR and v-qPCR results, for either treated or untreated biofilms. That means that all cells embedded to the biofilm matrix were metabolically active, infected or not. This result suggests that the application of phages in the biofilm environment does not impair the physiological state of all cells, but rather acts directly by killing and eliminating the most accessible cells. When 24 h old *P. aeruginosa* PAO1 biofilms were exposed to LUZ7, the number of adhered bacteria decreased by approximately 1.5 Log (Figure 6A) and with 14.1, the number decreased by 1.2 Log equivalent CFU/cm^2^ of the coupon (Figure 7A).The exposure of 24 h old D1 contamination led to a reduction of 0.72 Log with LUZ7 (Figure 6B) and a reduction of 1.7 Log equivalent CFU/cm^2^ of coupon with 14.1 (Figure 7B).

Collectively, LUZ7 treatment was the most effective against the PAO1 strain compared to 14.1. This was in accordance with the short latent period and the high burst size of LUZ7 previously characterized on PAO1. On the contrary, phage 14.1 was the most effective on the 24 h old D1 biofilm. Comparing the effect of phages against free cells on biofilm removal, a more positive and rapid effect was observed in eliminating the bacterial load of broth culture than for biofilm. However, observations made over the course of experiment enabled us to make the hypothesis that phage repartition in the system could be an explanation. Indeed, the turbidity of the liquid mineral minimum medium around stainless steel coupons (INOX 316L) was totally different between the control jar without phages and the other jars (Figure 8). 

To track the distribution of viruses in the studied system, the amount of phage present in the medium around the coupons after treatment with 14.1 was determined, as well as the amount associated with the biofilm. In the assays carried out with phage 14.1, an infectious dose equivalent to 4.1 × 10^10^ PFU was added to each jar. After 14 h, 6.3 × 10^11^ PFU and 5 × 10^10^ PFU in suspension was counted for the PAO1 strain and D1, respectively. This fraction only represents the phages which, at the end of treatment, were still completely free and virulent in the medium, since during the extraction protocol, part of the viruses adsorbed on the bacterial receptors was probably eliminated within the bacterial cell debris. The results also revealed the existence of a phage population adhered to the biofilm (10^8^ PFU). This population was lower than that found in the liquid medium, although it may have been underestimated if we consider the negative impact of sonication on the viability of phage particles highlighted by Melo et al. [77] and estimated in this study at 1 Log. LUZ7 phages showed similar behavior to a larger proportion of free-flowing phages in the medium than to associated phages (Table 9).

This unequal distribution of phages between the liquid and solid phases is in line with the spectrophotometry results, which demonstrated, during tests with LUZ7 phages, a very significant lysis of the planktonic population in the surrounding medium (Figure 8). In treated jars, negative values confirmed the absence of planktonic bacteria. On the basis of these observations, it is possible to discuss the mechanisms of action of bacteriophages by suggesting that:-Planktonic phages exerted lytic action on the free cells detached from the biofilm as a priority, allowing them to multiply in the mineral minimum medium around the coupons.-Some of these phages also adhered to the biofilm, thereby interfering with the development, as well as the spread of the biofilm in the surrounding environment.

Among the parameters likely to influence the efficacy of phages, the infectious dose used for the treatment is an important factor. By quantification of the 24 h biofilm cells, the MOI was estimated at 50 for the PAO1 strain and between 50 and 300 for the D1 strain in the case of the LUZ7 assays. These concentrations, which were much higher than those applied on the planktonic cell cultures (MOI 0.1; 1; 10), revealing the complexity of an attack targeting the biofilm. In the literature, Hanlon et al. [78] also demonstrated the greater susceptibility of the suspended cells to phages to the detriment of the adhered cells. In their experiments, an MOI of 100 resulted in a 1-log reduction in the viable cell number of the biofilm, and the log reduction factor increased to 2 with an MOI of 1000, without the age of the biofilm mattering. Indeed, the architecture and composition of the biofilm do not facilitate access to phages [78,79]. The superposition of several layers of cells causes an unequal distribution of nutrients and oxygen and promotes the appearance of bacterial sub-populations whose metabolism is changed. These modifications may also impact the expression of certain genes and thus the concentration and/or the presence of the receptors used by phages during infection on the cells’ surfaces [80,81].

Another parameter concerns the environment-promoting cells adhesion and biofilm’s growth, as well as the kind of surface present. Phages which multiply well under in vitro conditions can fail to replicate during treatment in vivo [25,69]. This means that all systems should be investigated before considering phages as potential green biocides. Garbe et al. [82] indeed showed a different behavior of the PAO1 strain cultivated in LB medium and in artificial sputum medium. Phage JG024 (*Myoviridae*, PB1-Like) was therefore less effective on biofilm. In the study of Scarascia et al. [38], three bacteriophages were further applied against a *P. aeruginosa*-enriched artificial seawater biofilm to validate their activity in saline condition. One previous study also highlighted that the ability of environmental *P. aeruginosa* isolates to form biofilm is correlated with the minerality of water without disturbing the metabolic activity rate [42]. Low minerality favors biofilm production. The use of mineral minimum medium, in our study, which was composed of mineral water with a lower minerality, could thus contribute to phage progression being limited through a high-density matrix but was well representative of the contaminations found on surfaces in contact with water. The ability of *P. aeruginosa* isolates to form biofilm has also been linked to the synthesis of pyocianin (PCN) [42], a virulence factor which can generate reactive oxygen species such as H_2_O_2_. Some authors have suggested that it is responsible for phage inactivation [25], but this requires further studies according the phages used.

## 5. Conclusions

Biofilms from *P. aeruginosa* on surfaces can barely be penetrated by current antimicrobial agents. The use of host-specific bacteriophages could be a potential alternative that could minimize the negative impacts of treatment on materials. This study enabled us to select specific lytic bacteriophages of environmental water network *P. aeruginosa* isolates. Bacteria were cultivated in a mineral minimum medium providing more realistic conditions than standard culture media. This medium was also found to be adequate for phage amplification and conservation over a long period of time. Among the nine phages studied, 14.1, B1 and LUZ7 were able to lyse up to 90% of the planktonic cells of PAO1 and D1 bacterial isolates. On 24 h old biofilms developing on stainless steel coupons, the maximum rate of biofilm elimination reached up to 1.7 Log after 14 h of treatment. Phages were found to be very efficient at limiting the spread of biofilm in the surrounding environment. Their effectiveness on attached cells was lower than on planktonic cells, probably due to the complex architecture of biofilm and the plasticity of bacteria cells. However, the real issue is the apparition of bacterial resistance when viruses and their planktonic hosts are challenged over long periods of time. Hence, in future works, it would be interesting to assess higher MOIs or phage cocktails or to combine their activity with some other biological molecules, without adding chemical products. Overall, the use of phages as biocontrol agents requires the development of safe, stable, and suitable solutions with a high antibacterial efficacy.

## Figures and Tables

**Figure 1 viruses-11-00749-f001:**
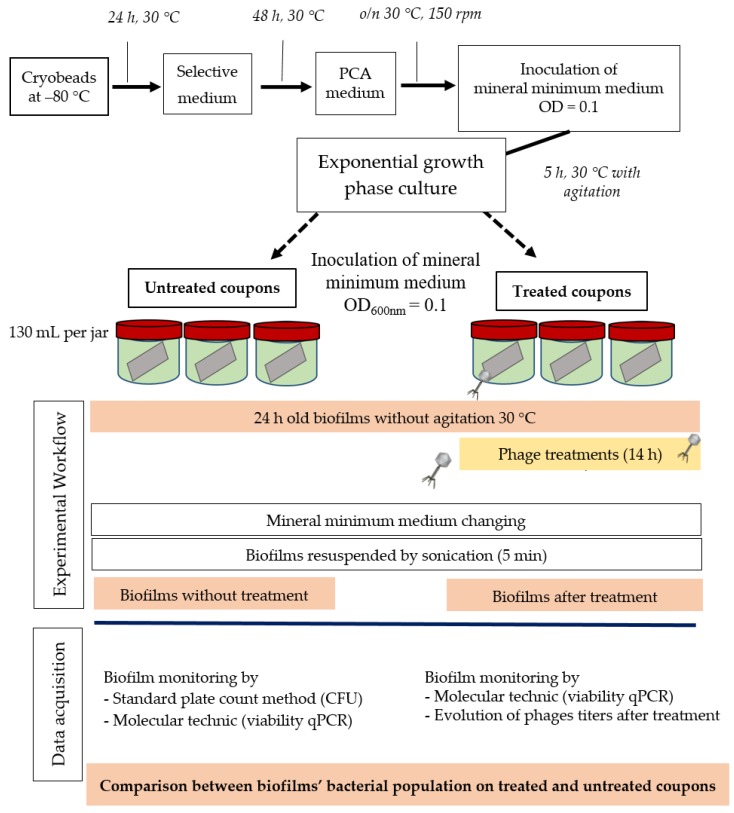
Efficiency of viruses’ treatment on *P. aeruginosa* cells from 24 h old biofilm: Experimental and analytical workflow steps. PCA: plate count agar. OD: optical density. CFU: colony forming unit. qPCR: quantitative polymerase chain reaction.

**Figure 2 viruses-11-00749-f002:**
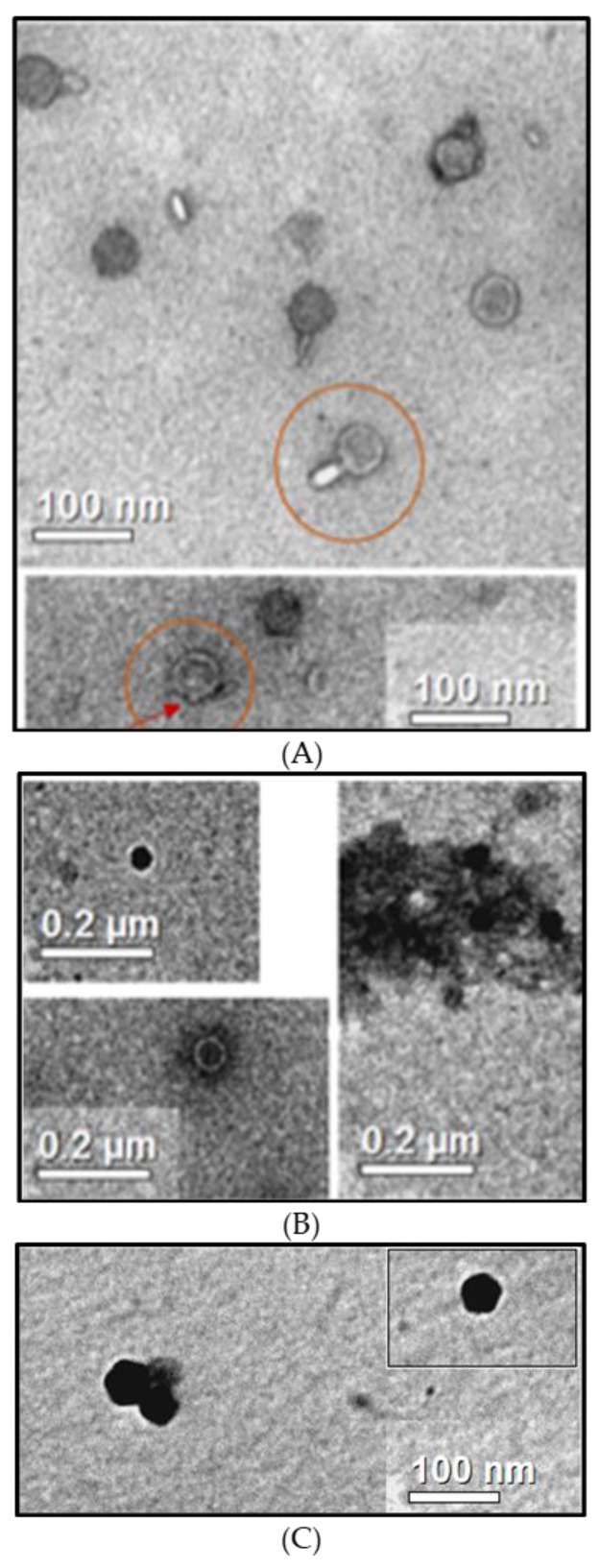
Transmission electron micrographs of phages 14.1 (**A**), LUZ7 (**B**), and B1 (**C**) stained with uranyLess. The circles show long tailed-particles and the arrow indicates the outgrowth replacing the original appendix lost by the vortexing step, limiting phage aggregate formation. The black color is probably due to positive staining.

**Figure 3 viruses-11-00749-f003:**
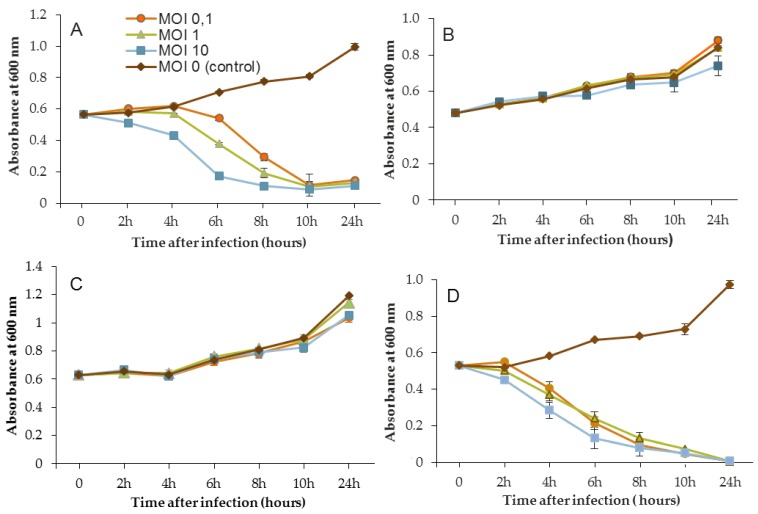
Activity of the phage LUZ7 against planktonic cells of four different *P. aeruginosa* strains (**A**) D1, (**B**) MLM, (**C**) ST395E, (**D**) PAO1. Each experiment was performed in triplicate, and bars indicate standard deviations. Multiplicity of infection (MOI).

**Figure 4 viruses-11-00749-f004:**
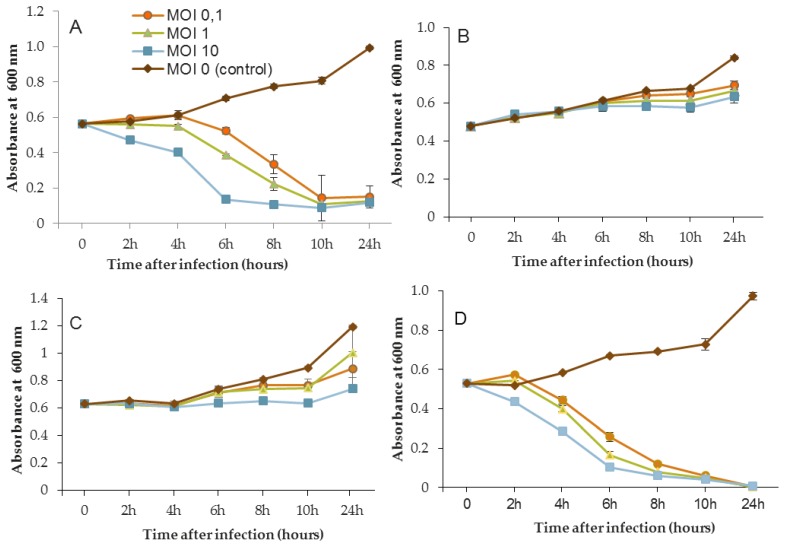
Activity of the phage 14.1 against planktonic cells of four different *P. aeruginosa* strains (**A**) D1, (**B**) MLM, (**C**) ST395E, (**D**) PAO1. Each experiment was performed in triplicate, and bars indicate standard deviations. Multiplicity of infection (MOI).

**Figure 5 viruses-11-00749-f005:**
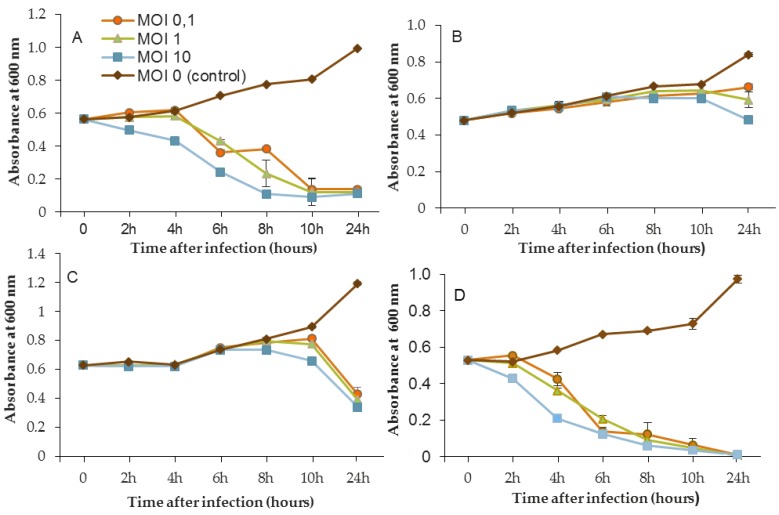
Activity of the phage B1 against planktonic cells of four different *P. aeruginosa* strains (**A**) D1, (**B**) MLM, (**C**) ST395E, (**D**) PAO1. Each experiment was performed in triplicate, and bars indicate standard deviations. Multiplicity of infection (MOI).

**Figure 6 viruses-11-00749-f006:**
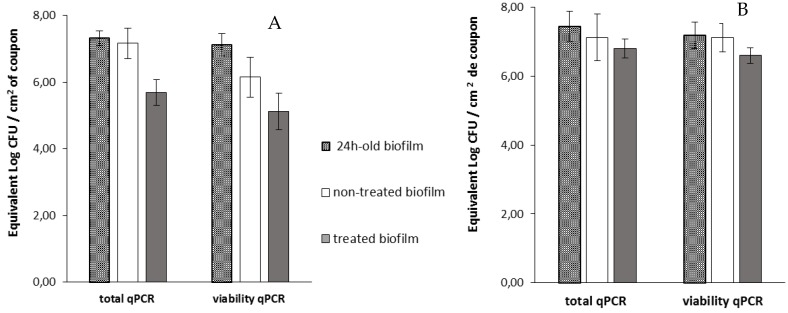
The effect of the bacteriophage LUZ7 on 24 h old biofilms of *P. aeruginosa* (**A**) on the PAO1 reference strain and (**B**) on D1 isolates from the thermal environment. The data represent the average of three biological replicates and the standard deviations are indicated by vertical bars.

**Figure 7 viruses-11-00749-f007:**
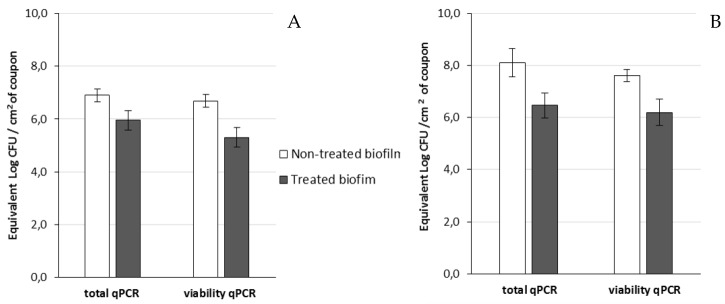
Effect of the bacteriophage 14.1 on 24 h old biofilms of *P. aeruginosa* (**A**) on the PAO1 reference strain and (**B**) on D1 isolates from the thermal environment. The data represent the average of three biological replicates and the standard deviations are indicated by vertical bars.

**Figure 8 viruses-11-00749-f008:**
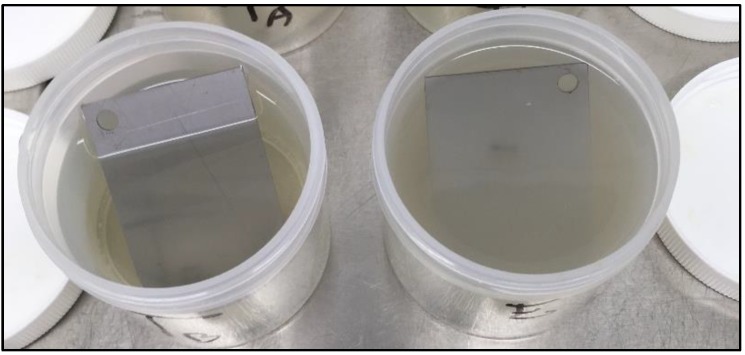
Overview of jars after 14 h of exposure to 14.1. On the left is one of the triplicate jars containing viruses. On the right is one of the triplicate controls without treatment applied. The turbidity of each one was compared by the optical density measurement of the surrounding mineral minimum medium (OD_600 nm_). Values were negative in the case of treated jars and between 0.2 and 0.4 for controls. Note that the effect of phages on the surrounding mineral minimum medium was only measured in the case of 14.1.

**Table 1 viruses-11-00749-t001:** Environnemental and clinical *P. aeruginosa* strains.

Name	Origin	Original Location	Reference
MLM	H_1	Tap water—faucet of consultation room	[42]
ST395E	H_3	Tap water—ICU faucet	[42,43]
ST395P	H_3	Clinical strain—ICU patient under ventilated assistance	[42,43]
D1	ThI_1	Thermal water	[42]
D2	ThI_1	Thermal water	[42]
E1	ThI_2	Thermal water	[42]
E6	ThI_2	Tap water—water network flushing	[42]
ST111	H_2	Tap water	[42]
PAO1	-	Reference strain isolated from a wound in 1954	[41]

ThI: thermal institute, H: hospital, ICU: intensive care unit.

**Table 2 viruses-11-00749-t002:** Phylogeny and molecular characteristics of the phages used.

Name	EL	LKD16	PEV2	LUZ7	14.1	LUZ19	MP22
Family	*Myoviridae*	*Podoviridae*	*Podoviridae*	*Podoviridae*	*Myoviridae*	*Podoviridae*	*Siphoviridae*
Genome	ds DNA						
Receptor	no data	type IV pili	unknown	AlgC (LPS)	LPS	type IV pili	type IV pili
Host	**PA14**	**PAO1**					**PAO1/PA14**
Sequence	**Yes**	**Yes**					**Yes**
Size	211215 bp	43200 bp	72697 bp	74901 bp	66235 bp	43548 bp	36409 bp
Topology	circular	linear	linear	linear	linear	linear	linear
Ref.	[49]	[44,45,46,47]				[50,51]	[52]

**Table 3 viruses-11-00749-t003:** Parameters of qPCR reagents and cycles.

	Kit PCR TaqMan^®^ QuantiNova Probe (Qiagen)		
	Sequences	Con.	Vol.
Mix	-		12.5 µL
Forward Primer (10 µM)	CCT-GAC-CAT-CCG-TCG-CCA-CAA-C (22nt)	0.4 µM	
Reverse Primer (10 µM)	CGC-AGC-AGG-ATG-CCG-ACG-CC (20nt)	0.4 µM	
TaqMan^®^ Probe (10 µM)	**6FAM-**GGT-CTG-GGA-ACA-GGT-CTA-CCA-CCA-CGG-**BHQ** (27nt)	0.2 µM	
DNA (Sample)	-		5 µL
RNAse-free water	-		5 µL
Total Volume	-		25 µL
Taq activation	2 min at 95 °C		
Cycles number	40		
Denaturation	5 s at 95 °C		
Hybridation/elongation	30 s at 60 °C		

**Table 4 viruses-11-00749-t004:** Host range of individual phages.

			*P.aeruginosa* Strains								
Taxonomy	Receptor	Phage	ST111	MLM	E1	E6	D1	D2	ST395E	ST395P	PAO1
*M*	*unknown*	EL	0/3	2/3	0/3	0/3	3/3	2/3	2/3	0/3	3/3
*P*	pili type IV	LDK16	0/3	2/3	0/3	0/3	3/3	1/3	2/3	0/3	3/3
*P*	*unknown*	PEV2	0/3	3/3	0/3	1/3	3/3	0/3	3/3	0/3	3/3
*P*	AlgC (LPS)	LUZ7	0/3	3/3	1/3	0/3	3/3	3/3	3/3	0/3	3/3
*P*	type IV pili	LUZ19	0/3	2/3	0/3	0/3	2/3	2/3	2/3	0/3	3/3
*M*	LPS	14.1	1/3	3/3	0/3	0/3	3/3	3/3	3/3	0/3	3/3
*S*	pili type IV	MP22	0/3	2/3	0/3	0/3	2/3	0/3	2/3	0/3	3/3
*unknown*		B1	0/3	3/3	0/3	0/3	3/3	1/3	3/3	0/3	3/3
*unknown*		LR1	0/3	1/3	0/3	0/3	1/3	0/3	1/3	0/3	1/3

M: *Myoviridae*; P: *Podoviridae*; S: *Siphoviridae.*
**x**/3: **x** represents the number of positive assay results obtained for the three replicates.

**Table 5 viruses-11-00749-t005:**
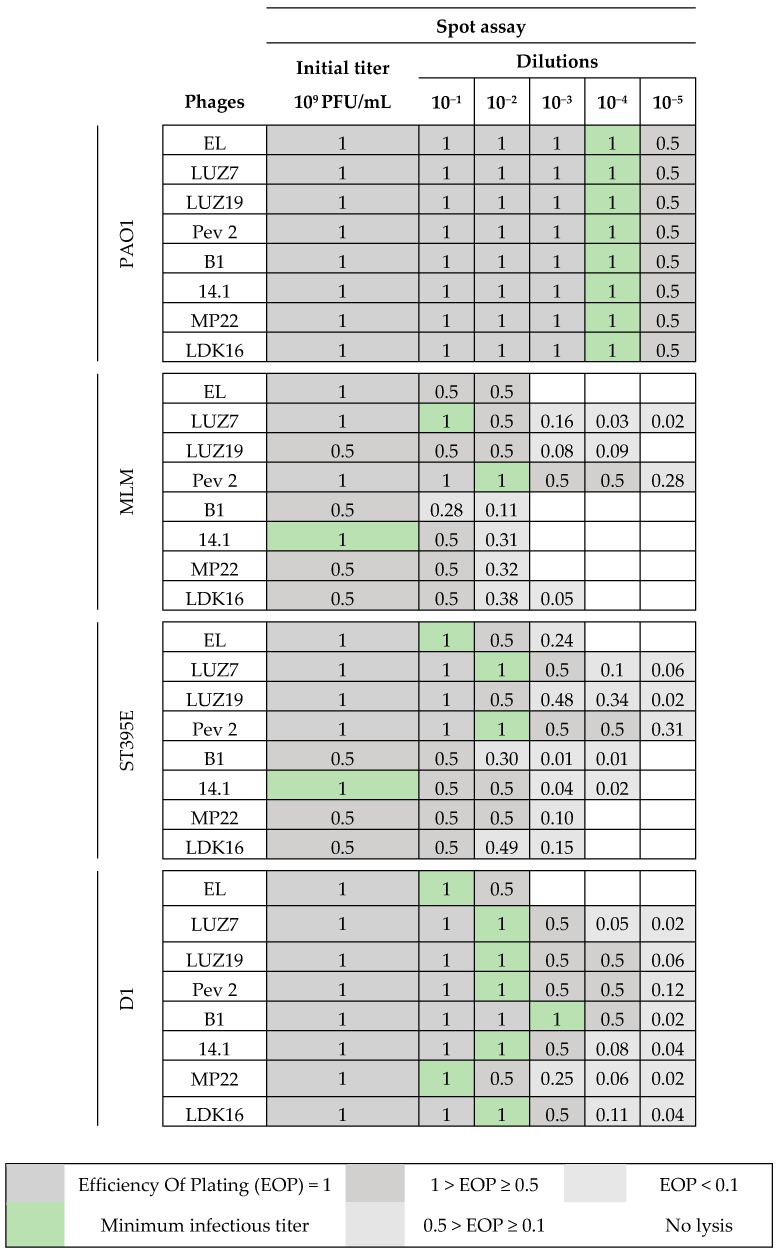
Spot assay results. Phage deposits were performed onto the double layer through serial dilutions and EOP values calculated with the PAO1 strain as reference.

**Table 6 viruses-11-00749-t006:** General features of viruses.

Name	Family	Latent Period	Burst Size (pfu/cell)
14.1	*Myoviridae*	20/25 min	70 pfu
LUZ7	*Podoviridae*	10 min	160 pfu
B1	*Podoviridae **	20 min	79/90 pfu

* based on transmission electron microscopy (TEM) observations.

**Table 7 viruses-11-00749-t007:** Relative cell quantification of 24 h old biofilms.

		PAO1 Strain	D1 Strain
**24 h Old Biofilms**	**qPCR ***	7.31 ± 0.23	7.44 ± 0.4
	**v-qPCR ***	7.12 ± 0.33	7.18 ± 0.39
	**SPC ***	7.40	8.20

* expressed as equivalent Log CFU/cm^2^ of coupon (v-qPCR and qPCR) and as Log CFU/cm^2^ for SPC. Viability qPCR (v-qPCR).

**Table 8 viruses-11-00749-t008:** Relative cell quantification of 24 h old biofilms untreated and treated with bacteriophages.

		Untreated	Treated **	Untreated	Treated **
		Phage LUZ7		Phage 14.1	
**PAO1 Strain**	**qPCR ***	7.16 ± 0.46	5.69 ± 0.39	6.90 ± 0.25	5.95 ± 0.37
	**v-qPCR ***	6.15 ± 0.60	5.11 ± 0.55	6.68 ± 0.24	5.30 ± 0.84
**D1 Strain**	**qPCR ***	7.12 ± 0.68	6.81 ± 0.27	8.1 ± 0.55	6.5 ± 0.47
	**v-qPCR ***	7.13 ± 0.41	6.60 ± 0.22	7.6 ± 0.21	6.2 ± 0.51

* Expressed as equivalent Log CFU/cm^2^ of the coupon. ** 24 h old biofilms treated by viruses for 14 h. Viability qPCR (v-qPCR).

**Table 9 viruses-11-00749-t009:** Particle enumeration of phages during experiments.

	LUZ7 Count (PFU)		14.1 Count (PFU)	
**Initial Dose Added**	6.4 × 10^10^		4.1 × 10^10^	
	planktonic	biofilm-associated	planktonic	biofilm-associated
**PAO1 Strain**	1.1 × 10^11^	5.5 × 10^7^	6.3 × 10^11^	2.1 × 10^8^
**D1 Strain**	3.41 × 10^9^	1.4 × 10^8^	5 × 10^10^	5.9 × 10^8^

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
