# Peer review of "Selection of Bacteriophages to Control In Vitro 24 h Old Biofilm of Pseudomonas aeruginosa Isolated from Drinking and Thermal Water"

_viruses, 2019, doi:10.3390/v11080749_

Round 1

Reviewer 1 Report

Major comments:

Overall, the experiments were well planned and the results the authors showed in this manuscript is interesting and important.

In this study, nine phages were tested against nine strains of P. aeruginosa.

I understand the authors tried to add explanatory comments with references, sentence by sentence in results and discussion. But, it is very tough for us readers to get the scientific viewpoints in this study. 

I prefer a conventional style to separate the discussion from the result section. Otherwise, it is not easy to catch the meaning of the conclusion. 

Minor comments:

1)  Please fix P. aeruginosa "PA01 " to "PAO1" in Table 1, Table 4, and Page 8, line 280.

2) In Table 4,  "PEV2"  should be "Pev2"

3) Page 15, line 492, and Page 16, line 500: " UFC"?  should be "CFU"?

4) Page 8, line 283: The word "v-qPCR" is suddenly used without explanation. 

Reviewer 2 Report

Authors of the manuscript "Selection of bacteriophages to control....." describe basically the comparison of efficacy of three out of 9 tested phages in the lysis of planctonic cells and decrease of biofilm of four Pseudomonas aeruginosa strains. The major novelty is the use of minimal mineral medium for their tests, which is different from typical conditions of P. aeruginosa phage activity testing  and could mimic conditions that occur in environmental water networks, spas etc. As such the manuscript provides a value information about the potential and efficacy of phages to  control P. aeruginosa under conditions similar to those tested by the authors. However, the English of the manuscript should be extensively corrected. In its present form the manuscript is not suitable for publication. My additional comments to be considered in the revised version of the manuscript are below.

L. 58: Replace "to an other biocides" with "to other biocides"

L 60: Replace "shed light bacteriophages" with "shed light on bacteriophages"

L. 83: Replace "VBNC" with "viable but non-culturable (VBNC)"

L. 141: Replace "solution" with "suspension"

L. 163-164: Efficiency of phage infection cannot explain why certain phages are more virulent than others on the same strain.

L. 169: Categorizing < 6 PFU as (-) is misleading. It suggests that there is no infection, whereas in fact it does mean the low efficiency of infection. The same applies to the data in Table 4 and Table 5. Why not to provide the average number of plaques in place of one -.

L.222-224: Rewrite the sentences for clarity.

L. 235-236: Authors should provide the results of control sonication of phage suspension of known titer to make sure that there is no decrease in the phage titer upon sonication.

L. 267: Italicize "gyrB"

Fig. 1A: The authors should provide the results of spot tests showing single plaques rather than the lysis zone. The latter is typically obtained with high lysate concentrations and may be due to bacteriocins that are present in the lysate or to lysis from without. Complete lysis with concentrated lysates cannot be considered as an evidence for the productive phage infection.

Certain references in the list should be corrected for consistency. Latin names should be italicized.

Reviewer 3 Report

General comments

The manuscript submitted by Magin et al., is an research article in which the selection of phages to control P. aeruginosa biofilm in vitro has been described. In my opinion, the topic of this manuscript is interesting, and may have practical implications in phage application in e.g. industry. The manuscript structure is long because of detailed description in material and methods section and results, but this feature do not decrease of its value. The language and style of the manuscript are clear. The structure of the manuscript is diversified, it contains tables and figures which are helpful and summarize the obtained results. Methods that were used in the experiments are appropriate.

The suggested changes are minor and I recommend the article to be published after minor corrections.

The suggested corrections:

(1)   Title: I would like to suggest change in the title of the manuscript (e.g. “Selection of bacteriophages to control 24h – old biofilm of Pseudomonas aeruginosa in vitro”).

(2)           Abstract: page 1 ,line 11: authors should introduce the abbreviation Gram negative ( as Gram-) when used by the first time.

(3)           Page 2, line 83: could the authors explain what VBNC is?

(4)   Page 4, line 131: Why the authors wrote here “propagate” instead of :amplify”?

(5)   Page 5, line 167-169: Could the authors provide reference based on which the lysis intensity was evaluated?

(6)   Page 7, Figure 2, the authors used “DO”. Did they mean OD (optical density).

(7)   Page 11, line: 390-392, the authors should be more careful and put spaces where it is needed.

(8)   Conclusion: In my opinion this section should be shortened and rephrased.

Round 2

Reviewer 2 Report

The current version of the manuscript is improved as compared to its previous version. The additional results provided by the authors show plaques as requested (L. 174-181, and Figures A1-5). An important information, that is missing concerns the specification of the strain that was used for the preparation of lysates to study the infectivity of particular phages to particular strains with the use of diluted lysates. Was it one strain that was used to propagate all phages or lysates to test were prepared in cells of the same strains as the tested strain in each case. The ability of a given phage to productively infect a given strain may depend on a strain used for the propagation of this phage. Thus, such information should be provided. Additionally, in the summary of results in Table 5 the authors should use the efficiency of plating (EOP) value in place of confusing statements of the kind "The plaques were considered as completely lysed (+++), partially lysed (++-) etc." Plaques cannot be partially lysed. They may be clear or turbid and this should be stated by the authors.  But knowing the number of plaques on a layer of each tested strain with a given phage dilution the authors can easily calculate EOP and provide the EOP values for each phage using a selected strain (e.g. PAO1) as a reference. In other case the summary of spot assay results in Table 5 is unclear.

Minor comments:

L. 66: Replace "achieved" with "performed"

L. 114, 115, 116, 341, 355, 356, 367, 379, 380, 383, 384, 394, 465, 609, 826, Table 2, Table 6,  : Italicize:  "Caudovirales", "Myoviridae", "Podoviridae", "Siphoviridae"

L. 227, 237, 248, 415, 417, 439, 443, 445, 451, 456, 457, 526, 532, 533, 535, 581: Lines  with these numbers contain the statement "(Error. Reference source not found.)". Either these statements should be removed or appropriate citations should be included in the text of the manuscript where they are required.

L. 244: Replace "the amount" with "the number of phages"

L. 279: Italicize "P. aeruginosa"

L. 279: What do the authors mean by "Efficiency of viruses on P. aeruginosa biofilm cells" They should be more precise.

L. 336, 337: Replace "Gram-bacteria" with "Gram-negative bacteria"

L. 341 and elsewhere in the text: Italicize taxonomic names

The manuscipt contains typographical errors, which should be corrected, e.g., L. 398, 641. Taxonomic names of bacterial species in the reference list should be corrected, so that the second part of each name does not start from a capital letter.
